# Perinuclear Anti-Neutrophil Cytoplasmic Antibodies (pANCA) Impair Neutrophil Candidacidal Activity and Are Increased in the Cellular Fraction of Vaginal Samples from Women with Vulvovaginal Candidiasis

**DOI:** 10.3390/jof6040225

**Published:** 2020-10-16

**Authors:** Andrea Ardizzoni, Arianna Sala, Bruna Colombari, Lavinia Beatrice Giva, Claudio Cermelli, Samuele Peppoloni, Anna Vecchiarelli, Elena Roselletti, Elisabetta Blasi, Robert T. Wheeler, Eva Pericolini

**Affiliations:** 1Department of Surgical, Medical, Dental and Morphological Sciences with Interest in Transplant, Oncological and Regenerative Medicine, University of Modena and Reggio, 41125 Emilia, Modena, Italy; andrea.ardizzoni@unimore.it (A.A.); arianna.sala@unimore.it (A.S.); bruna.colombari@unimore.it (B.C.); claudio.cermelli@unimore.it (C.C.); samuele.peppoloni@unimore.it (S.P.); elisabetta.blasi@unimore.it (E.B.); 2Graduate School of Microbiology and Virology, University of Modena and Reggio, 41225 Emilia, Modena, Italy; lavinia.giva@gmail.com; 3Department of Medicine, University of Perugia, 06132 Perugia, Italy; vecchiar10@gmail.com (A.V.); E.Roselletti@exeter.ac.uk (E.R.); 4Department of Molecular and Biomedical Sciences, University of Maine, Orono, ME 04469, USA; 5Graduate School of Biomedical Sciences and Engineering, University of Maine, Orono, ME 04469, USA

**Keywords:** *Candida*, VVC, pANCA, ASCA, CAGTA

## Abstract

Vulvovaginal candidiasis (VVC) is primarily caused by *Candida albicans* and affects 75% of childbearing age women. Although *C. albicans* can colonize asymptomatically, disease is associated with an increased *Candida* burden, a loss of epithelial tolerance and a breakdown in vaginal microbiota homeostasis. VVC symptoms have been ascribed to a powerful inflammatory response associated with the infiltration of non-protective neutrophils (PMN). Here, we compared the immunological characteristics of vaginal fluids and cellular protein extracts obtained from 28 VVC women and from 23 healthy women colonized by *Candida* spp. We measured the levels of antibodies against fungal antigens and human autoantigens (anti-*Saccharomyces cerevisiae* antibodies (ASCA), *C. albicans* germ tube antibodies (CAGTAs) and perinuclear anti-neutrophil cytoplasmic antibodies (pANCA)), in addition to other immunological markers. Our results show that the pANCA levels detected in the cellular protein extracts from the vaginal fluids of symptomatic women were significantly higher than those obtained from healthy colonized women. Consistent with a potential physiologically relevant role for this pANCA, we found that specific anti-myeloperoxidase antibodies could completely neutralize the *ex vivo* killing capacity of polymorphonuclear cells. Collectively, this preliminary study suggests for the first time that pANCA are found in the pathogenic vaginal environment and can promptly impair neutrophil function against *Candida*, potentially preventing a protective response.

## 1. Introduction

*Candida* is a common component of the vaginal microbiota in healthy women and causes vulvovaginal candidiasis (VVC), an opportunistic fungal infection caused by several *Candida* spp. but, in particular, *C. albicans* [1]. This condition affects 75% of healthy women during their reproductive age, at least once in their lifetime. An additional 5–10% of women suffer from recurrent VVC (RVVC), which is characterized by four or more episodes of VVC per year [2]. The symptoms of both VVC and RVVC include itching, burning and redness of the vaginal mucosa with white vaginal discharge [2]. Although rodent infection models are useful for studying the human condition, they also present several limitations. For instance, unlike humans, laboratory rodents do not naturally harbor *C. albicans* as a commensal; moreover, in contrast to humans, the vaginal pH of mice is neutral [3,4].

*C. albicans* is considered to be an immunoreactive pathogen in the context of VVC [3]. Current research is consistent with the hypothesis that a threshold of *C. albicans* burden breaks tolerance and allows for the onset of VVC, mediated by the immune response. Unfortunately, the inflammation triggered by *C. albicans*–epithelial interactions elicits neutrophil recruitment, through several mechanisms including S100A8 release, but does not result in clearance of the pathogen. In fact, while neutrophils are ineffective against *C. albicans*, they are believed to actually exacerbate symptoms. The lack of neutrophil anti-*C. albicans* activity may be due to heparan sulfate, co-colonizing bacteria, fungal factors, or a combination of all three [5,6]. While there is a dearth of human studies to corroborate the mechanistic research performed with mouse disease models, it is believed that recruited neutrophils contribute to host damage and disease rather than protection against the fungus.

The loss of immunological tolerance to commensal microbiota/mycobiota-derived molecules and disruption of the intestinal mucosal barrier are associated with other host-driven inflammatory diseases such as intestinal bowel disease, Crohn’s disease and colitis [7,8,9]. In these diseases, immunogenic stimuli promote the production of anti-*Saccharomyces cerevisiae* antibodies (ASCA) and perinuclear anti-neutrophil cytoplasmic antibodies (pANCA) [10,11]. The presence of ASCA has not been linked to any direct immune activity, while pANCA are known to be associated with neutrophil extracellular traps (NETs), which contain their target, myeloperoxidase [12]. Furthermore, pANCA can directly activate neutrophils *ex vivo* to produce damaging reactive oxygen species [13] and are believed to play, clinically, an active role in vasculitis [14].

Considering the hypothesized immunoreactive nature of VVC and the belief that VVC arises when the fungal burden rises above a certain threshold, it is intriguing that we have seen high levels of extracellular DNA (eDNA)—consistent with NET presence—in the vaginal environment of infected, but not colonized, women [15]. This led us to hypothesize that VVC causes an increased production of pANCA, which could contribute to the inflammatory and damaging immune environment during VVC. We therefore measured the levels of several antibodies associated with inflammatory mucosal diseases, leading to the discovery that pANCA levels are high in the cellular fraction of vaginal samples from symptomatic women. Biological assays of pANCA activity confirmed their capacity to promote neutrophil reactive oxygen species (ROS) production but, to our surprise, also revealed their ability to block neutrophil candidacidal activity. These promising results, although obtained from a small cohort of women, suggest that pANCA elicitation may be an important marker for VVC that contributes both to the local damaging inflammatory response and to the inability of vaginal neutrophils to control the *Candida* burden during VVC.

## 2. Materials and Methods

### 2.1. Ethics Statement

All women signed informed consent forms in accordance with the Declaration of Helsinki. Local Ethical Committee CEAS (Comitato Etico delle Aziende Sanitarie, Umbria, Italy) approval was received for the whole study (VAG1 n. 2652/15 on 1 February 2016 and subsequent amendments of 28 November 2016 and 15 June 2017). The killing activity and the oxidative burst experiments were performed with human blood-derived neutrophils. The blood samples were provided from the Immuno-Transfusional hospital service of the Policlinico di Modena and were collected from healthy donors.

### 2.2. Subjects

This study includes 28 women colonized by *C. albicans* with VVC symptoms (indicated as “VVC”), 23 asymptomatic *Candida*-colonized women (*C. albicans* and non-*albicans Candida* spp.) (indicated as “colonized”) and 9 healthy non-colonized women. All the patients enrolled in the present study are non-diabetic and 19 to 53 years old and attended the microbiological diagnostic service of the University Hospital “Santa Maria della Misericordia”, in Perugia (Italy), over the period February 2016 to August 2017. Enrollment, clinical parameter determination and obtaining consent were carried out as previously described [15,16]. As detailed in Appendix A, a first group of patient samples (yellow coded) was assessed for antibodies and S100A8 levels in vaginal fluids (VF) only. Following this first set of results, the sampling was extended to include the preservation of the cell pellets, which allowed the testing of protein extracts from cellular fractions (PE) from samples from a second set of participants (blue coded; Appendix A). Prior to enrollment, each woman was asked to answer a questionnaire indicating their health status and current symptoms of vaginal disease. None of the patients was taking any treatment for symptoms at the time of enrollment. A case of VVC due to *Candida* was defined as (1) *Candida* isolation from the vaginal swab and (2) the presence of at least two among the following signs and symptoms: vaginal discharge, itching, burning and/or dyspareunia. None of the recruited women had RVVC, as indicated by the absence of documented or subject-reported, repeated VVC episodes per year.

### 2.3. Neutrophil Infiltration Scoring, pH Analysis and Bacterial Identification

Vaginal swabs were taken and immersed in 1 mL of saline, as previously described [16]. The pH value was tested in some vaginal samples by means of pH-Fix strips (Machery-Nagel GmbH & Co. KG, Germany). The morphotypic evaluation of vaginal microbiota was performed by the observation of a fresh preparation of vaginal fluid or by the Gram staining method under a light microscope. In particular, the presence or absence of Lactobacilli was verified. After sampling, the vaginal swab was directly plated on (a) Tryptic Soy Agar II with 5% sheep blood, (b) Gardnerella Selective Agar with 5% human blood or (c) Columbia Agar with 5% Sheep Blood (all from Becton Dickinson) to test for vaginal colonization by Lactobacilli, *G. vaginalis* or *S. agalactiae* (Group B Streptococci, GBS), respectively. Then, to confirm Lactobacilli, *G. vaginalis* or *S. agalactiae* identification, mass-spectrometry (MALDI-TOF) analysis using a VITEK MS system (Biomérieux S.A., 1 Rue Lénine, 94200 Ivry-sur-Seine France) was performed. Samples were examined under a light microscope (Olympus, Milan, Italy) to evaluate the presence of polymorphonuclear cells (PMNs) after staining with the Papanicolaou technique [16,17]. The PMNs were counted in four fields at ×400 magnification, and their number was expressed as the average number of PMN/field. The PMN were scored on a scale from 0 to 2, as previously described [16]. Briefly, for 0 PMN/field, the score was 0; for 1 to 10 PMN/field, the score was 1; with 11 to 40 PMN/field, the score was 2. This method allowed the semi-quantitative analysis of neutrophil numbers. *Candida*-positive vaginal samples were categorized as no or low neutrophil infiltration (LI; score ≤ 1) or high neutrophil infiltration (HI; score = 2). This cohort included 28 women who were *C. albicans*-positive with high neutrophil infiltration (HI), 18 women who were *C. albicans*-positive with low neutrophil infiltration (LI) and 5 women with non-*albicans Candida* (NAC) infection with no or low neutrophil infiltration (LI).

### 2.4. Sample Collection and Species Identification

A separate vaginal swab was taken from each subject and immersed in 2 mL of phosphate buffered saline (PBS). After sampling, the vaginal swab was plated on CHROMagarTM Candida (VWR International p.b.i., Milan, Italy). The plates were kept at 37 °C for 48 h to evaluate the vaginal colonization by *Candida*. The presence of *Candida* was confirmed by MALDI-TOF using the VITEK MS system (Biomérieux S.A., France). Subsequently, the vaginal fluid was centrifuged at 3000 relative centrifugal force (rcf) for 5 min. Vaginal fluids were recovered, aliquoted, frozen at −80 °C and then used for ASCA, pANCA, S100A8 and *C. albicans* germ tube antibody (CAGTA) determination. The cellular fraction in the pellet from a subset of samples (Appendix A) was resuspended and divided into 2 aliquots. One aliquot was used for protein extraction, and the other aliquot was frozen in TRIzol reagent (Thermo Fisher Scientific, Monza, Italy) for the assessment of gene expression. Due to variability in the amounts of sample collected, some vaginal samples were not able to be tested for all parameters.

### 2.5. Determination of ASCA, pANCA and S100A8 in Vaginal Fluids

One hundred microliters of vaginal fluids from each sample were used for the following tests: (1) the determination of IgG and IgA antibodies against mannan from *Saccharomyces cerevisiae* in human body fluids (ASCA IgG/IgA); (2) the assessment of human perinuclear anti-neutrophil cytoplasmic antibodies (pANCA); (3) the evaluation of S100 Calcium Binding Protein A8 (S100A8). All these tests were performed by means of specific ELISA kits (all from MyBioSource, San Diego, CA, USA). According to the manufacturers’ instructions, ASCA IgG-/IgA-positive samples were considered as those with above 0.5 U/mL, and pANCA-positive samples were considered as those containing more than 100 pg/mL of the antibody. 

### 2.6. Determination of ASCA and pANCA in the Cellular Fraction

Cellular fractions obtained from a subset of vaginal samples were washed with PBS and lysed with mammalian protein extraction reagent in the presence of protease inhibitors (all obtained from Pierce, Rockford, IL, USA). The protein concentration of each cellular fraction was measured by using the bicinchoninic acid assay (Pierce). Equal amounts (25 μg) of total protein from each sample were suspended in 100 µL of PBS and used to analyze ASCA IgG/IgA (U/mL) and pANCA (pg/μg) concentrations with specific ELISA kits (MyBioSource), as above, according to the manufacturers’ instructions. The pANCA IgG index was calculated as follows: VF level (pg/mL) + [250 × PE level (pg/μg)].

### 2.7. Determination of CAGTA in Vaginal Fluids

One hundred microliters of vaginal fluids from each sample were used for the determination of the presence of CAGTA, by means of an indirect immunofluorescent assay kit (Vircell S.L., Granada, Spain). With this method, specific IgG antibodies raised against antigens specifically located on the surface of *Candida albicans* germ tube or hyphal cells (both provided in the kit) (CAGTA) can be detected in human body fluids. Such germ tube-specific antibodies can be detected by pre-absorbing the biological samples onto *C. albicans* yeast cells (provided in the kit heat inactivated) and then testing the pre-absorbed fluids using the CAGTA kit. The kit procedure was partly modified in order to detect the presence of IgA antibodies, by using a polyclonal anti-human IgA Alexa Fluor 555 conjugated antibody (goat, 1.035 mg/mL, dilution 1:100) (Jackson Immunoresearch, Cambridgeshire, CB7 4EX, United Kingdom) as the secondary antibody. The fluorescence of fungi was visualized by means of epifluorescence microscopy with a Nikon Eclipse 90i (Nikon Instruments, Tokyo, Japan). The magnification was 40X, and the scale-bar represents 50 μm. Operationally, we considered as CAGTA positive those samples that showed staining of at least 95% of the hyphae and CAGTA negative those samples that showed <5% of hyphal fluorescent staining.

### 2.8. Gene Expression Analysis

The cellular fractions from a subset of vaginal samples were lysed using TRIzol (Life Technologies, Monza, Italy). Total RNA was extracted and retro-transcribed by using the Moloney murine leukemia virus reverse transcriptase reaction (M-MLV RT), as described in the manufacturer’s instructions. The cDNA concentration was determined using a spectrophotometer. For each target gene (human GADPH, CD11b and MPO), primers were selected using the OligoAnalyzer 3.1 software. All the primers are listed in Appendix A. Real-time PCR (quantitative PCR) was performed in 96-well PCR plates (Thermo Scientific, Waltham, MA, USA) using SYBR Green (BioRad, Milan, Italy). For the real-time PCR reaction, 400 ng of cDNA was used. All samples were measured in triplicate. The expression levels of the CD11b and MPO genes in asymptomatic and symptomatic women were calculated by the formula 2 ^Ct mean (GADPH)^/2 ^Ct mean (gene)^. The amplification conditions were the same for the CD11b, MPO and GADPH genes: 3 min at 95 °C, 40 cycles of 10 s each at 95 °C and 30 s at the primer-specific temperature. The experiments were performed using an Applied Biosystems 7300 (Thermo Scientific).

### 2.9. Preparation of Polymorphonuclear Cells

Human peripheral blood neutrophils, obtained from healthy donors, were separated by density gradient centrifugation on a Ficoll-Hypaque (Euroclone, Milan, Italy), followed by the hypotonic lysis of erythrocytes. Neutrophils were then washed twice with PBS (Sigma Aldrich, St. Louis, MO, USA) and suspended at the desired working concentration.

### 2.10. Candida albicans ATCC Strain

*Candida albicans* SC5314 (ATCC MYA-2876) were used for killing activity and oxidative burst experiments. Fungal cultures were maintained by biweekly passages onto Sabouraud Dextrose Agar (SDA) plates (OXOID, Milano, Italy). The day before each experiment, fresh *Candida* cultures were seeded onto yeast-peptone-dextrose (YPD) plates and incubated at 37 °C. After overnight incubation, fungal cells were harvested with a sterile inoculating loop, suspended in phosphate-buffered saline (PBS, EuroClone, Whethereby, UK), washed twice by centrifugation at 3500 rpm for 10 min, counted in a Burker’s chamber and suspended at the desired concentration in RPMI 1640 medium supplemented with 10% heat-inactivated fetal bovine serum (hiFBS) (Defined Hyclone, Logan, UT, USA), gentamicin (50 mg/mL; Bio Whittaker, Verviers, Belgium), Ciproxin (2 mg/mL; ICN) and l-glutamine (2 mM; EuroClone, Milan, Italy).

### 2.11. Neutrophil Oxidative Burst

Human peripheral blood neutrophils (2 × 10^6^/mL, 50 μL/well) were seeded in a 96-well black-transparent plate (Perkin Elmer) in the presence of the medium or the anti-human myeloperoxidase Ab (pANCA) (USBiological, 50 μL/well) and with *C. albicans* (4 × 10^6^/mL, 50 μL/well); then, the MitoSox Red Probe (Thermo Fisher) (5 μM) was added in each well; hence, the fluorescence emission (ex/em 510/580) was measured kinetically every 5 min for 80 cycles using a fluorimeter (FluoroSkan, Thermo Fisher, Monza, Italy). An irrelevant IgG antibody (rabbit IgG, Jackson ImmunoResearch) was used as an isotype control. The commercially available pANCA antibodies have previously been shown to perform similarly to patient-isolated pANCA IgG [18,19,20].

### 2.12. Neutrophil Candidacidal Activity

The killing activity of neutrophils was determined by the colony-forming unit (CFU) inhibition assay. Briefly, neutrophils (10^5^ cells, in 0.05 mL of suspension/well) were incubated in the presence of the medium or the anti-human myeloperoxidase Ab (pANCA) (USBiological, 50 μL/well) in flat-bottom 96-well microtiter tissue culture plates, infected with 10^4^
*C. albicans* cells in 0.1 mL of RPMI plus 10% hiFBS, and incubated for 2 h at 37 °C under 5% CO_2_. An irrelevant IgG antibody (rabbit anti-human IgG, Jackson ImmunoResearch) was used as an isotype control. After incubation, the plates were vigorously shaken, and the cells were lysed by adding Triton X-100 (0.1% in distilled water; final concentration in the well, 0.01%). The samples were then spread on Sabouraud dextrose agar plus chloramphenicol (50 μg/mL) in triplicate, and the CFU values were evaluated after 48 h of incubation at 37 °C. The control cultures consisted of *C. albicans* incubated in RPMI 1640 plus 10% FCS without effector cells. Killing activity was expressed as the percentage of CFU inhibition according to the following formula: percentage of killing activity = 100 − (CFU experimental/CFU control) × 100.

### 2.13. Statistical Analysis

Statistical analyses were performed with Graphpad 8 (Prism). Quantitative variables were tested for normal distributions with the Shapiro–Wilk test. Statistical differences between groups were assessed by the two-tailed Student’s t-test when parametric; the statistical differences for CD11b gene expression, the pANCA IgG index and the number of PMN/field were assessed with the non-parametric Mann–Whitney U test. Pearson’s correlation test was used to analyze the data of Appendix A. Fisher’s exact test was used to compare VVC and healthy colonized women for clinical parameters and for CAGTA-IgA-positive samples. A value of *p* < 0.05 was considered significant. Significance throughout the figures is indicated with * *p* < 0.05, ** *p* < 0.01 and **** *p* < 0.0001.

## 3. Results

Vulvovaginal candidiasis, in contrast to *Candida* colonization, is associated with a strong infiltration of neutrophils, extracellular DNA and exposed glucan epitopes [15]. Recently, inflammation and neutrophil extracellular traps have been found to be associated with the production of autoantibodies that can alter neutrophil function [12,14,21]. To test if VVC occurs in the context of autoantibody-mediated immunomodulation, we recruited both healthy (*Candida*-colonized) and symptomatic patients (VVC) and collected vaginal swabs from both groups. While all the symptomatic patients in our cohort had *C. albicans*, the asymptomatic women were colonized by *C. albicans* (78%) or by non-*albicans Candida* spp. (NAC: 22%; *C. glabrata*, *C. krusei* or *C. guillermondi*). Unlike asymptomatic women, who had little-to-no neutrophil infiltration, symptomatic women had high neutrophil infiltration (Figure 1A); this finding correlated with enhanced levels of CD11b expression (Figure 1B) but was not accompanied by increased levels of myeloperoxidase (Figure 1C) or S100A8 (Table 1 and Appendix A, Figure 1D). The strong association of symptoms with neutrophilic involvement is consistent with previous work by us and others [3,15,22].

Given the hypothesized, but still unproven, associations between altered microbiota and symptomatic disease, we compared the two cohorts of women for the presence of specific bacteria. In both groups, Lactobacilli were found in most of the samples, and there was no significant difference in their frequency between symptomatic and asymptomatic women (*p* = 1.0, Fisher’s exact test), consistent with an unchanged pH, ranging between 4.4 and 4.6 (Table 2 and Appendix A). Additionally, there was no correlation between the lack of Lactobacilli, or the presence of two pathobionts (*Streptococcus agalactiae* [GBS] and *Gardnerella vaginalis*), and the occurrence of VVC (Table 2 and Appendix A; *p* = 1.0, Fisher’s exact test). These data are in line with previous work by us and other groups, suggesting that there is no relevant synergy or antagonism between these bacterial spp. and *C. albicans*, at least with respect to VVC [15,23,24,25].

The strong correlation between high neutrophil infiltration and symptomatic disease suggested that other immune responses might also be enhanced. Previous work has described humoral immune responses against *C. albicans* in other tissues and in VVC [26,27], and we therefore sought to determine the levels of anti-*Candida* antibodies in symptomatic and healthy women. We first tested for the presence of broadly reacting anti-*Saccharomyces cerevisiae* antibodies (ASCA), which, despite their name, bind to mannan and therefore to taxonomically distinct fungi [11]. Both IgA and IgG ASCA were assessed in whole vaginal fluids (VF) from all the participants. These antibodies were also assessed in a second group of samples, for which both VF and protein extracts from the cellular fraction (PE) were available, as detailed in the Materials and Methods section, in Table 1 and in Appendix A. As shown in Appendix A, in VF, the number of ASCA-IgA-positive samples was low (cut-off = 0.5 U/mL, as set by the manufacturer). For women with *C. albicans*, ASCA IgA were found in 8/28 samples (29%) in the group of symptomatic women and in 2/18 samples (11%) in the group of asymptomatic women, which indicated a trend but was not a significant difference (*p* = 0.2736, Fisher’s exact test). Asymptomatic women with NAC had a similar rate of ASCA-IgA-positive samples (1/5, namely, 20%). Additionally, the ASCA IgA samples from symptomatic and asymptomatic women had similar levels of antibodies (Table 1 and Appendix A). In contrast to IgA levels, the ASCA IgG levels were undetectable for all but one sample, which was barely over the cut-off level of 0.5 U/mL (Appendix A). ASCA IgA and IgG were not found in the PEs from any of the samples analyzed.

Despite the nearly universally low levels of ASCA antibodies, we hypothesized that there might be an increase in the *Candida*-specific humoral response in symptomatic patients. To test this, we assayed for the presence of *Candida albicans* germ tube antibodies (CAGTA), which are known to be associated with *Candida* infection [28]; such antibodies are much more specific than the more promiscuous ASCA. The CAGTA assay is an all-or-none assay with a binary readout. When measuring the CAGTA IgA, we detected a significantly higher frequency of positive samples in symptomatic women (19/28 samples; 68%) as compared to asymptomatic *C. albicans*-colonized women (6/18 samples; 33%) (Appendix A, Figure 2A; *p* = 0.0340, Fisher’s exact test). Interestingly, 4 out of 5 samples from the NAC-colonized asymptomatic women were CAGTA IgA positive (Appendix A and Figure 2A,B).

The enhanced prevalence of anti-*Candida* antibodies in VVC patients argues for the occurrence of a more pronounced pathogen-specific local humoral immune response in symptomatic infections. Previous work, including our own, suggests an association between neutrophilic infiltration, extracellular DNA and symptomatic infection [15,22]. Because neutrophil activation and NETs are associated with the stimulation of autoantibodies such as pANCA [12,14], we assessed whether these autoantibodies were detectable in the vaginal swabs from symptomatic and/or asymptomatic women. To our surprise, pANCA IgG were found above the detection limit (100 pg/mL) in most of the VF samples analyzed, i.e., in 23/28 (82%) of the samples from symptomatic women and in 19/23 (83%) of the samples from asymptomatic women (Appendix A; difference not significant *p* > 0.05). Such pANCA levels in the VF trended higher but were not significantly different in symptomatic versus asymptomatic women (Figure 3A; *p* = 0.8037). Since pANCA IgG are directed against neutrophil myeloperoxidase [12], which is associated with the cellular fraction that would include NETs, their presence was also evaluated in the protein extract (PE) of the cellular fraction, in a subset of vaginal samples, for which both VF and PE had been collected and stored from VVC and colonized asymptomatic women, as detailed in the Materials and Methods section and in Appendix A. Interestingly, the pANCA levels detected in the PE from the symptomatic women were significantly higher than those from the asymptomatic colonized women, irrespective of the *Candida* spp. involved (Appendix A and Figure 3B). Similarly, total pANCA levels (pANCA IgG index = VF plus PE) were also slightly higher in symptomatic as compared to asymptomatic women (Figure 3C; *p* = 0.0496). Consistent with the lack of a significant difference in the levels of pANCA in VF, there were no significant correlations between pANCA levels in VF and PE identified in symptomatic patients (Appendix A), while a slight but significant positive correlation was observed in asymptomatic controls (Appendix A). Taken together, these data suggest that elevated levels of cell-associated pANCA are related to VVC and raise the possibility that pANCA may have a role in the symptoms.

Samples from a small cohort of nine healthy non-colonized women (Appendix A) show that the levels of pANCA in VF do not differ from VVC patients and healthy colonized women. However, while we attempted to collect data from PE, we found that there was a vanishingly small cell pellet from healthy non-colonized women such that we were not able to extract enough protein for analysis. Therefore, it was not possible to measure the levels of pANCA in PE in such samples. 

Neutrophils are hypothesized to play a host-damaging role in VVC, which is paradoxically associated with a diminished ability to kill *Candida* [3,6,29]. Interestingly, pANCA can alter neutrophil physiology by enhancing their activation and reactive oxygen species (ROS) production, which likely drives vasculitis [12,14]. Currently, it is unknown if pANCA may also affect neutrophil-mediated antimicrobial activity. Thus, we carried out in vitro experiments to test the effect of pANCA on ROS production and test the capacity of neutrophils to kill *Candida* in the presence of pANCA. Using an ex vivo challenge model, we showed that peripheral blood neutrophils exposed to pANCA produced more ROS but also lost their capacity to kill *Candida* (Figure 4). This reduced killing capacity was not due to a general loss of neutrophil function during the short *Candida*-neutrophil killing assay, because pANCA did not block the ability of neutrophils to produce an oxidative burst in response to *Candida* stimulation. In particular, pANCA caused the expected direct stimulation of ROS and also enhanced ROS production in the context of *Candida* challenge (Figure 4B and B-inset).

## 4. Discussion

A central paradox about vulvovaginal candidiasis is the presence of both strong neutrophilic infiltrates and a high fungal burden [3,29]. It is believed that the symptoms associated with VVC are largely due to neutrophil-coordinated inflammation, a situation in which the recruited neutrophils damage the host but have a limited capacity to kill the pathogen in the vaginal environment [3,29]. This is reminiscent of diseases such as chronic granulomatous disease, in which innate immune cells recruited during pulmonary fungal infection are hyperinflammatory but fail to contain the infection [30]. Based on a comparison with other hyperinflammatory mucosal conditions, such as Crohn’s disease and inflammatory bowel disease (IBD), which are associated with antibodies against microbial and self-antigens [31], we measured several humoral immune parameters of the vaginal environment, focusing on *Candida*-colonized and VVC-symptomatic women, our aim being to identify correlates of an overactive antibody response to *Candida* during VVC. We found increases in several antibodies during VVC, including *Candida*-specific CAGTA antibodies and autoreactive pANCA antibodies. The correlation of pANCA with VVC is especially interesting in light of the hyperinflammation associated with VVC, because these antibodies activate neutrophils to produce ROS. We have found that, *in vitro*, pANCA hyperactivate neutrophils to make ROS, as expected, but intriguingly, such antibodies also completely abrogate the immune cells’ ability to kill *Candida*. Taken together, these data suggest a new disease-associated mechanism in VVC whereby an infection-associated increase in pANCA levels both exacerbates host damage, via ROS production, and reduces the ability of infiltrating neutrophils to contain the infection, because of their impaired killing ability.

The hyperactivation of host humoral immunity, with the production of both autoreactive and commensal microbe-directed antibodies, is associated with most autoimmune conditions, including overactive mucosal inflammatory diseases such as Crohn’s disease [32,33]. Similar to IBD, it is currently believed that VVC is linked to the loss of epithelial tolerance to *Candida* colonization [3]. Even though there is a constant interplay between innate and adaptive immunity to maintain the homeostasis of the vaginal mucosa, to date, little is known on the role of antibodies in the immunological tolerance occurring during VVC. We have found that mannan-specific ASCA antibodies, often used as a diagnostic for IBD, are not robustly associated with VVC. Nonetheless, the relatively small size of our sample makes it difficult to exclude clinical significance for this parameter. By contrast, we saw a significantly enhanced production of *Candida* germ tube-specific IgA antibodies in VVC patients, implying the occurrence of such a specific humoral response to the fungus in the vaginal mucosa. 

Similarly to the ASCA marker, the serological analysis of host-directed pANCA levels is a diagnostic tool for patients suffering from chronic intestinal inflammation as well as vasculitis [33,34]. Interestingly, analysis of the cellular fraction from a limited number of our patient samples revealed a significant increase in pANCA levels in VVC women as compared to colonized ones. When we attempted to obtain data also from the cohort of healthy non-colonized women, we realized that only a very small cell pellet could be collected from these samples, so we were not able to extract enough protein for analysis. Although these are negative data, the lack of cell-associated pANCA is nonetheless consistent with the hypothesis that pANCA plays a role in the pathogenesis of VVC. In addition, by merging the data obtained from the analysis of both cellular and fluid fractions, we strengthen the occurrence of significantly higher levels of pANCA during VVC. Further studies enrolling a larger number of samples are warranted to establish whether the presence of pANCA in the cellular fraction may be potentially useful as a specific biomarker of lost epithelial immune tolerance at the vaginal level.

Previous work, associating autoimmune vasculitis and increased levels of pANCA, has pointed to an important role of such antibodies in directly activating neutrophils to produce ROS and thereby enhance tissue damage [35]. We confirmed the activity of pANCA in directly activating neutrophil ROS production, demonstrating that this is even further enhanced by concomitant challenge with *Candida*. This is consistent with the high neutrophilic infiltrate and the epithelial damage observed in VVC [3,22,29]. We tested if there was any physiologically relevant change to the neutrophil–*Candida* interaction, and we found that pANCA-stimulated neutrophils are completely abrogated in their ability to kill *Candida ex vivo*. This is remarkable, first, because it is consistent with the paradox of high neutrophilic infiltration but a failure to contain the fungal burden. Furthermore, it is consistent with other work that demonstrated the anergy of vaginally isolated neutrophils in a murine infection model [29]. Future studies will be carried out to determine if autoantibodies have an effect on vaginally isolated neutrophils from a murine VVC model similar to the effect they had on human peripheral neutrophils in our study. The higher ROS production caused by pANCA stimulation and the extracellular traps previously observed in vaginal samples suggest that pANCA may enhance neutrophil-mediated epithelial damage, although work in a murine model suggests that host damage is more closely linked to fungal burden than neutrophil infiltration [29]. Although technically difficult to prove and beyond the scope of the present study, our data lead to the speculation that neutrophils from patients with VVC may be decorated with pANCA, thus lacking candidacidal activity.

Our study assessed if there was an association between VVC and a number of other factors, including *Lactobacillus* colonization, bacterial co-infections, increased myeloperoxidase activity and neutrophil anergy [29,36,37]. There was no association of VVC with the absence of Lactobacilli or presence of other bacteria, which is in line with other data arguing against a role for bacteria in VVC [23,24,38]. Although high levels of neutrophils have been detected in the vaginal samples of VVC women, we did not find increases in myeloperoxidase activity or S100A8. Previous reports have identified both high levels of S100A8 and strong neutrophil recruitment in a murine infection model [6,29,39]. It is unclear why we did not obtain a similar result in our clinical samples. We cannot exclude that such a difference may be due to the different human *vs* murine species considered, but this could be due to the lower expression of these genes during the response to vaginitis. Nonetheless, the *CD11b* levels were high in our VVC samples and correlated with the high neutrophilic infiltrate detected in human VVC. As for the lack of correlation between the levels of S100A8 and PMN recruitment, it is likely that in such a complex system, other cytokines/chemokines together with fungal factors may contribute to the outcome of the inflammation.

Collectively, our results, although preliminary and obtained with a small cohort of women, demonstrate for the first time that pANCA increase in the vaginal environment during VVC and that such anti-MPO antibodies drastically impair neutrophils’ anti-*Candida* effector function, which would prevent their protective role against the infection. In addition, the enhanced levels of pANCA in VVC may indicate a loss of immunological tolerance at the vaginal mucosa, which would ultimately lead to VVC onset. It will be of considerable interest to extend these findings to larger patient cohorts in the future.

## Figures and Tables

**Figure 1 jof-06-00225-f001:**
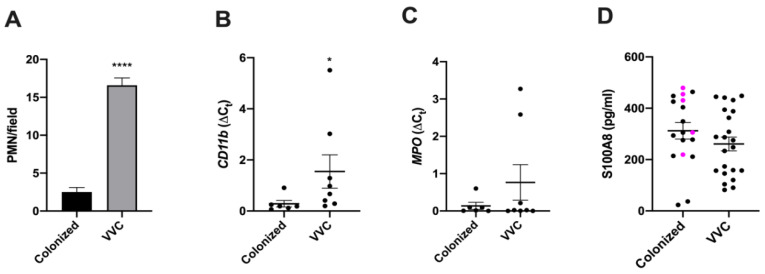
PMN number, *CD11b* and *MPO* gene expression and S100A8 determination. (**A**) Mean ± SEM of PMN number/field from vulvovaginal candidiasis (VVC) or colonized women. (**B**,**C**) Mean ± SEM of the relative expression of CD11b (**B**) and MPO (**C**) genes from cDNA obtained from cellular pellets of VVC and colonized women (data were obtained from the analysis of samples indicated in Appendix A). (**D**) S100A8 was assessed in vaginal fluids (VF) of VVC and colonized women. The samples from women colonized by non-*albicans Candida* (NAC) species are colored in magenta. Level of significance is indicated by **** *p* < 0.0001 and by * *p* = 0.0127.

**Figure 2 jof-06-00225-f002:**
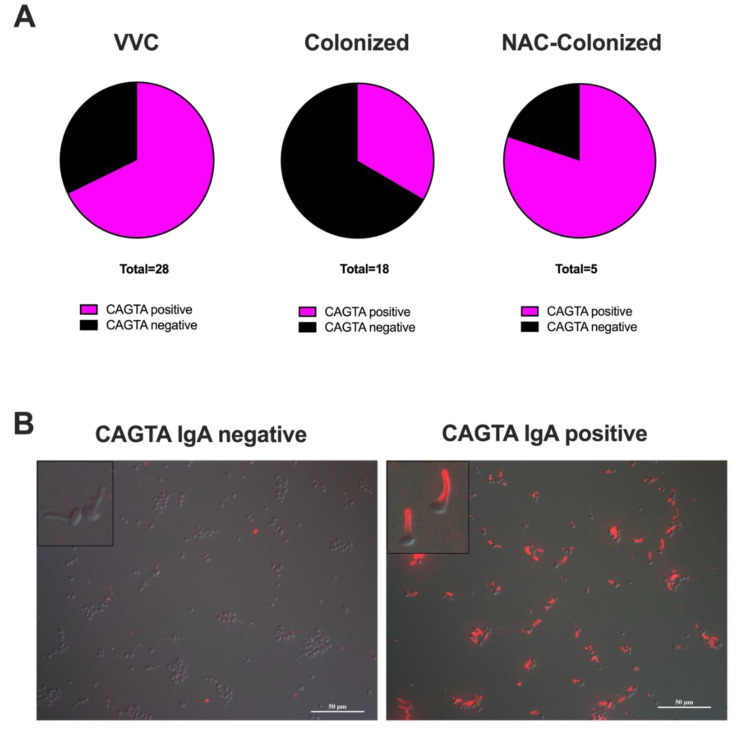
*C. albicans* germ tube antibody (CAGTA) IgA determination. (**A**): Pie charts of CAGTA IgA in positive (magenta) and negative (black) samples from VF of VVC women and *C. albicans*- or NAC-colonized women. (**B**): Representative images from samples with no CAGTA IgA (colonized women: CAGTA IgA negative, SP-8605) and with CAGTA IgA (VVC women: CAGTA IgA positive, SP-4900). The *Candida* cells shown are used to capture specific CAGTA and are part of the analysis kit. Magnification = 40X; scale bar = 50 μm.

**Figure 3 jof-06-00225-f003:**
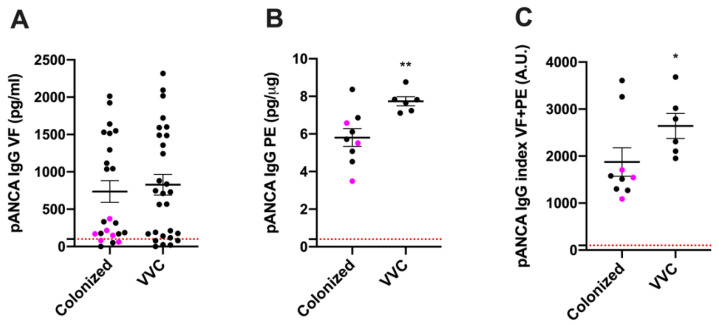
pANCA levels in VF and PE. (**A**,**B**) Mean ± SEM of pANCA IgG (pg/mL) in vaginal fluids (VF) and in protein extracts (pg/μg) (PE). (**C**) pANCA IgG index in VF + PE expressed as arbitrary units (A.U.) from VVC and colonized women. The samples from women colonized by NAC species are colored in magenta. Levels of significance are indicated by ** *p* < 0.01 and * *p* = 0.0496.

**Figure 4 jof-06-00225-f004:**
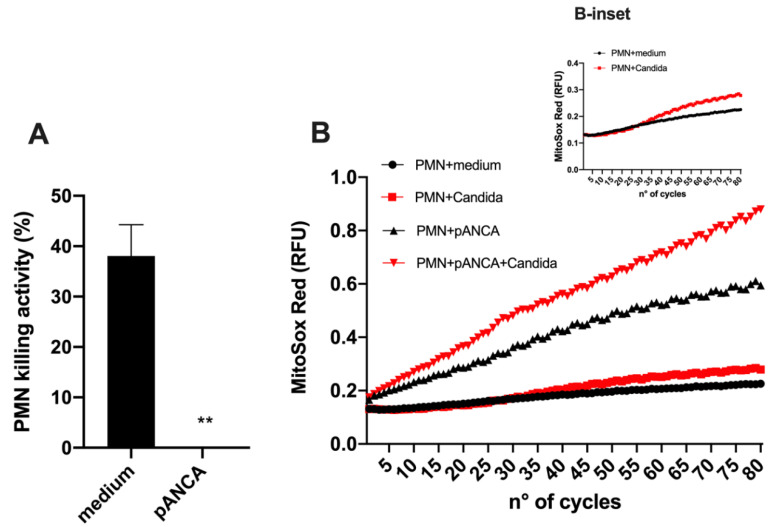
Candidacidal activity. (**A**) Mean ± SEM of the percentage killing activity of human neutrophils (PMN) against *C. albicans* SC5314 in the presence or absence of pANCA. Significance is indicated with ** *p* < 0.01. (**B**) Oxidative burst of neutrophils in the presence or absence of pANCA was kinetically analyzed (for 80 cycles, where 1 cycle was 5 min) after in vitro exposure to *C. albicans* (SC5314). B-inset highlights the oxidative burst of PMN against *C. albicans* SC5314 (already shown in Panel B) without pANCA treatment. Results are the means of two independent experiments with duplicate samples.

**Table 1 jof-06-00225-t001:** Summary of the data (mean ± SEM) on anti-*Saccharomyces cerevisiae* antibodies (ASCA), anti-neutrophil cytoplasmic antibodies (pANCA) and S100A8 determination in vaginal fluids and protein extracts.

Clinical Condition	PMN Infiltration	Fungal Species	ASCA in VF (U/mL)		pANCA IgG		S100A8 (pg/mL)
			IgA	IgG	VF (pg/mL)	PE (pg/μg)	
**VVC**	**High or Massive (HI)**	*C. albicans*	0.60 ± 0.05	0 ± 0	827.53 ± 26.11	7.74 ** ± 0.24	260.81 ± 27.07
**Colonized**	**Low or None (LI)**	*C. albicans* and NAC	0.34 ± 0.18	0.02 ± 0.02	736.77 ± 145.05	5.80 ± 0.47	312.03 ± 32.18
**Table key:**	Data are expressed as the mean ± SEM			VF: vaginal fluids		PE: protein extract	

** means *p* < 0.01.

**Table 2 jof-06-00225-t002:** Summary of the microbiological features of the vaginal samples assessed.

Clinical Condition	PMN Infiltration	No. of Samples	Fungal Species	Bacteria	Lactobacilli	Co-Infection with GBS	Co-Infection with *G. vaginalis*	pH (Mean ± SEM)
VVC	High or massive (HI)	28	*C. albicans*	14/14	23/28	4/14	3/14	4.6 ± 0.1
Colonized	Low or none (LI)	18	*C. albicans*	10/10	15/18	2/10	0/10	4.4 ± 0.1
	Low or none (LI)	5	*C.* non-*albicans*	5/5	5/5	1/5	0/5	4.6 ± 0.2

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
