# Peer review of "Perinuclear Anti-Neutrophil Cytoplasmic Antibodies (pANCA) Impair Neutrophil Candidacidal Activity and Are Increased in the Cellular Fraction of Vaginal Samples from Women with Vulvovaginal Candidiasis"

_jof, 2020, doi:10.3390/jof6040225_

Round 1

Reviewer 1 Report

The terminology seems odd. Women who suffer from VVC are colonized as well, the women addressed in this manuscript as colonized are asymptomatically colonized.

Why are the women asymptomatically colonized with NAC species not excluded? You are comparing a group with C. albicans infected women to a group of women asymptomatically colonized with both C. albicans and NAC species. Please explain.

How do the authors explain the fact that there is no difference in s100A8 between groups when the PMN number is lower in asymptomatic infection. What about other cytokines?

The Fidel lab described PMN dysfunction in VVC. Your PMN killing assays are performed with HPBN's. Is there a way to test this with PMNs isolated from vaginal fluid? They might respond differently.

Author Response

Reviewer 1

The terminology seems odd. Women who suffer from VVC are colonized as well, the women addressed in this manuscript as colonized are asymptomatically colonized.

Response: Yes, we of course agree with your comment. In the Materials and Methods section of the revised manuscript, we specify that all the women are colonized by Candida (page 3, lines 93-95), some of them asymptomatically and others with symptoms of VVC. However, to keep it simple, in this revised version we maintain in the figures and tables the term “colonized” to indicate asymptomatically colonized women and the term “VVC” to indicate women colonized with symptoms.

Why are the women asymptomatically colonized with NAC species not excluded? You are comparing a group with C. albicans infected women to a group of women asymptomatically colonized with both C. albicans and NAC species. Please explain.

Response: As described in Materials and Methods, we enrolled all patients with VVC, regardless of whether it was C. albicans or NAC. We included all species of Candida because it is reported in the literature (Kennedy MA and Sobel JD, PMID: 21308556) that VVC can be caused also by NAC species. We agree with the reviewer that it appears that we excluded symptomatic NAC-colonized women from the cohort. However, the lack of symptomatic VVC patients with NAC is only by chance, because all the women colonized by NAC species belonged to the asymptomatically colonized group. Excluding the asymptomatic women colonized by NAC does not materially change the overall analysis. 

How do the authors explain the fact that there is no difference in s100A8 between groups when the PMN number is lower in asymptomatic infection. What about other cytokines?

Response: We are well aware that in VVC the interplay of cytokines/chemokines is very complex (especially in humans) and that the role of S100A8 in the PMN recruitment is just one part of a very complex picture. This parameter may not be enough per se to indicate a difference in PMN recruitment, and it is possible that S100A8 expression is significantly lower in PMNs from the symptomatic patients. Since the most interesting findings from this work were focused on autoantibodies, we are planning to recruit a larger cohort of women in order to better correlate this autoantibody response with the pattern of a larger group of cytokines/chemokines secretion. It will be interesting to see if this unexpected lack of correlation between S100A8 and neutrophil number is seen in this next cohort. In the revised manuscript we have added a sentence in Discussion (page 12, lines 444-446) to address this lack of correlation and to better specify that other cytokines/chemokines may play a primary role in the inflammation during VVC.

The Fidel lab described PMN dysfunction in VVC. Your PMN killing assays are performed with HPBN's. Is there a way to test this with PMNs isolated from vaginal fluid? They might respond differently.

Response: This is a very good point and it will be very interesting to continue studying both human VVC patients and mouse models of VVC to determine how neutrophil infiltrates fail to control Candida burden in VVC. Of course, the work from the Fidel and Noverr laboratories has focused on the mouse model of VVC, where it is possible to isolate neutrophils with vaginal lavage without the same level of ethical concern for the health of the female patient. Unfortunately, it is not possible at this point to bridge this ethical concern in our Institutional Review Board so we are not able to answer this interesting question. We agree that peripheral and vaginal neutrophils may very well respond differently, especially in the context of asymptomatic Candida colonization or symptomatic VVC. We have added text to the Discussion section (page 11, lines 423-426) to include this potential complication.

Reviewer 2 Report

well written and interesting article. Recommended for publication

Author Response

Reviewer 2

well written and interesting article. Recommended for publication

Response: Thank you for your kind comments.

Reviewer 3 Report

This manuscript presents promising preliminary results that  pANCA antibodies are observed in  patients with VVC. The major conclusion is that the presence of these antibodies may account for the ineffective response of neutrophils seen in patients with VVC.  However, pANCA antibodies are also found in colonized patients - so it seems like data from the appropriate controls are not included.  If the results from women without Candida were similar to women with Candida colonization, then the conclusion could be made that pANCA antibodies contribute to pathogenesis in VVC patients. The authors should indicate that these are preliminary/ promising since the study reflects a very small group of patients.

Materials and Methods/ Results

  • Lines 9-96 The explanations of abbreviations need to be included the first time the abbreviation is mentioned. This should be checked throughout the manuscript. For instance VF and PE are used on lines 95 and 96 but not explained until lines 272-273.
  • Line 155. Germ tube antibody assay: The explanation of the assay needs to be clarified. The source of Candida for the assay (both yeast cells for adsorption and Candida with germ tubes; new hyphae) needs to be specified.   The legend in Figure 2 seems to imply that the Candida are from the patients?  
  • Line 199. Neutrophil candicidal activity assay. This assay is confusingly presented. Are the first and second paragraphs described one or two different assays. If the first paragraph is describing the ROS assay, this should be stated. The second paragraph does not mention antibody so assumedly the neutrophils are incubated as described in lines 201-202. Line 205 mentions that a irrelevant IgG antibody was used as an isotype control. When? The correct way the  assay should be performed is that neutrophils are either  incubated with pANCA or the isotype control. So it is not clear if this assay is valid without knowing this information.   
  • Sample collection. There is no mention in the manuscript of the number of Candida found in patients. It would be interesting to know if this correlated at all with neutrophil infiltrate.
  • Candida should be italicized throughout the manuscript
  • Make sure all tables and figures are correctly referred to.
    • Only S100A8 levels are referred to Table 1 but not the antibody levels in VF, PE. Instead only Table S1 is referred to.
    • Figure 2A is referred to but not Figure 2B (sometimes references to figures are Fig. and other times Figure- should be consistent)

Some minor comments

  • Line 69 - define eDNA
  • In the introduction, you should explain why S100A8 levels are determined.
  • There should be some mention that the results reflect a very small group of patients.
  • Line 154 – it is not similar to cases such as CGD. This is due to a known immunological defect.

Author Response

Reviewer 3

This manuscript presents promising preliminary results that  pANCA antibodies are observed in  patients with VVC. The major conclusion is that the presence of these antibodies may account for the ineffective response of neutrophils seen in patients with VVC.  However, pANCA antibodies are also found in colonized patients - so it seems like data from the appropriate controls are not included.  If the results from women without Candida were similar to women with Candida colonization, then the conclusion could be made that pANCA antibodies contribute to pathogenesis in VVC patients. The authors should indicate that these are preliminary/ promising since the study reflects a very small group of patients.

Response: We agree with the Reviewer that our results are promising preliminary experiments with a small patient cohort that link the in vitro inhibitory activity of pANCA antibodies to levels in vaginal samples from symptomatic vs colonized women. We have added text to the Abstract (page 1, lines 35-36), Introduction (page 2, line 79) and Discussion (page 12, line 447) to be absolutely clear about the preliminary nature of this study. The Reviewer asks for controls to examine pANCA levels from the cellular pellet of vaginal swabs from healthy non-colonized women. While we attempted to collect this data, we found that there was a vanishingly small cell pellet from healthy non-colonized women such that we were not able to extract enough protein for analysis. Because our main goal was to understand how Candida can be associated with symptomatic disease or asymptomatic colonization, we focused on these two cohorts of women. However, given the understandable interest of this Reviewer in the comparisons between healthy non-colonized women and Candida colonized/symptomatic VVC patients, we have now included the rest of the data collected on these samples (supplemental Table 3). Although this lack of data from non-existent cell pellets might be interpreted as a hole in our study, we believe that the lack of a cell pellet in healthy non-colonized women instead argues that there can’t be any cell pellet-associated pANCA in this group of women. Thus, while this is negative data, it is nonetheless consistent with the hypothesis that pANCA plays a role in the pathogenesis of VVC.

Materials and Methods/ Results

  • Lines 9-96 The explanations of abbreviations need to be included the first time the abbreviation is mentioned. This should be checked throughout the manuscript. For instance VF and PE are used on lines 95 and 96 but not explained until lines 272-273.

Response: We agree with the reviewer comment. Amended.

  • Line 155. Germ tube antibody assay: The explanation of the assay needs to be clarified. The source of Candida for the assay (both yeast cells for adsorption and Candida with germ tubes; new hyphae) needs to be specified.   The legend in Figure 2 seems to imply that the Candida are from the patients?  

Response: We agree with the reviewer comment. The Candida are from the assay kit and not from patients. The vaginal supernatants with/without CAGTA are from the patients. Amended (see Materials and methods section page 4, lines 166-168 and revised legend to Figure 2).

  • Line 199. Neutrophil candicidal activity assay. This assay is confusingly presented. Are the first and second paragraphs described one or two different assays. If the first paragraph is describing the ROS assay, this should be stated. The second paragraph does not mention antibody so assumedly the neutrophils are incubated as described in lines 201-202. Line 205 mentions that a irrelevant IgG antibody was used as an isotype control. When? The correct way the  assay should be performed is that neutrophils are either  incubated with pANCA or the isotype control. So it is not clear if this assay is valid without knowing this information.   

Response: We agree with the reviewer comment. The assay was conducted exactly as suggested by the reviewer. Amended (see Materials and methods section page 5, line 206 and lines 216-222).

  • Sample collection. There is no mention in the manuscript of the number of Candida found in patients. It would be interesting to know if this correlated at all with neutrophil infiltrate.

Response: The patients were all diagnosed clinically for colonization with Candida or for VVC, diagnoses that require a clinically relevant level of fungi found in vaginal samples. Beyond this binary score, we don’t have additional quantitative data on fungal burden. It is interesting that high fungal burden was not correlated with symptomatic infection in the only human VVC challenge study that has been published (Fidel et al.; PMCID: PMC387876). In this previous study, the neutrophil infiltrate did not correlate with fungal burden. However, this previous study had a relatively small cohort and further work to quantify both fungal burden and neutrophil infiltration in additional cohorts is warranted. We certainly plan to include quantification of both parameters in follow-up work.

  • Candida should be italicized throughout the manuscript

Response: Amended

  • Make sure all tables and figures are correctly referred to.

Response: Thank you, checked.

  • Only S100A8 levels are referred to Table 1 but not the antibody levels in VF, PE. Instead only Table S1 is referred to.

Response: Amended

  • Figure 2A is referred to but not Figure 2B (sometimes references to figures are Fig. and other times Figure- should be consistent)

Response: Amended

Some minor comments

  • Line 69 - define eDNA
  • In the introduction, you should explain why S100A8 levels are determined.
  • There should be some mention that the results reflect a very small group of patients.
  • Line 154 – it is not similar to cases such as CGD. This is due to a known immunological defect.

Response: Thank you. Amended.

Round 2

Reviewer 3 Report

The authors have done an excellent job in addressing the suggestions of the reviewers.